# Cytokines TNFα, IFNγ and IL-2 Are Responsible for Signal Transmission from the Innate Immunity Protein Tag7 (PGLYRP1) to Cytotoxic Effector Lymphocytes

**DOI:** 10.3390/cells9122602

**Published:** 2020-12-04

**Authors:** Tatiana N. Sharapova, Elena A. Romanova, Olga K. Ivanova, Lidia P. Sashchenko, Denis V. Yashin

**Affiliations:** Laboratory of Molecular Immunogenetics of Cancer, Institute of Gene Biology RAS, Vavilova 34/5, 111394 Moscow, Russia; elrom4@rambler.ru (E.A.R.); olga.k.ivanova@gmail.com (O.K.I.); sashchenko@genebiology.ru (L.P.S.); yashin_co@mail.ru (D.V.Y.)

**Keywords:** PGLYRP1, cancer immunology, monocytes, TREM-1, cytokines, programmed cell death

## Abstract

Studies on the mechanisms of activation of cytotoxic lymphocyte subpopulations are an important research direction in modern immunology. This study provides a detailed analysis of the effect of Tag7 (PGRP-S, PGLYRP1) on the development of lymphocyte subpopulations cytotoxic against MHC-negative tumor cells in a pool of peripheral blood mononuclear cells (PBMCs). The results show that Tag7 can bind to the TREM-1 receptor on the surfaces of monocytes, thereby triggering the expression of mRNA TNFα and IFNγ. The appearance of these cytokines in conditioned medium leads to IL-2 cytokine secretion by CD3^+^CD4^+^ lymphocytes. In turn, IL-2 facilitates unspecific activation of three cytotoxic cell subpopulations in the PBMC pool: NK (CD16^+^CD56^+^), CD3^+^CD4^+^ and CD3^+^CD8^+^. These subpopulations appear after a certain period of incubation with Tag7 and show toxicity against tumor cells.

## 1. Introduction

One of the main goals in molecular immunology is to understand the mechanisms of functioning of the human immune system. Studies on proteins involved in immune activity allow for deeper insight into how this system normally operates in healthy individuals and what factors are responsible for its disturbances characteristic of various pathological states. These studies also contribute to the development of novel means of immune therapy. The cooperative action of innate and adaptive immunity provides for defense against alien agents, and cells involved in the innate and adaptive immune response are stimulated by specific factors. These factors include the family of peptidoglycan recognition proteins (PGRPs) encoded by evolutionarily conserved genes. One of them is the Tag7 (PGLYRP1) protein, which is structurally conserved from insects to mammals [1]. Tag7 in insects is a component of the antimicrobial defense system, which is involved in signal transmission to the TLR4 receptor and thereby activates the production of antimicrobial peptides [2]. Mammalian Tag7 also contributes to antimicrobial defense, but the mechanism of this process has not yet been elucidated in detail [3].

According to our recent data, Tag7 is involved in the immune response against tumor cells. In particular, it can form an equimolar complex with heat shock protein Hsp70 and induce programmed cell death (apoptosis and necroptosis) in tumor cell lines by activating the TNFR1 receptor on the target cell membrane. It has also been shown that the Tag7–Hsp70 complex may be secreted by active cytotoxic CD3^+^CD8^+^ lymphocytes [4,5]. Tag7 also plays the role of surface co-receptor on CD3^+^CD4^+^ and can bind with Hsp70 on MHC-negative tumor cell surfaces. This interaction is critical for FasL–Fas signaling and tumor cell lysis [6].

The role of Tag7 in the elimination of tumor cells is not limited to the direct cytotoxic effect of its complex with Hsp70. It has been shown that Tag7 is capable of binding to the TREM-1 receptor on the monocyte surface and inducing the development of cytotoxic lymphocyte subpopulations in peripheral blood mononuclear cells (PBMCs) [7].

TREM-1 is an innate immunity receptor expressed on monocytes and neutrophils. It is known that this receptor operates synergistically with TLR, enhancing the immune response [8]. TREM-1 activation leads to the secretion of proinflammatory cytokines TNFα, IL-1β and IL-6, which activate immune cells [9]. Cytokines play a key role in regulating the immune response against various pathogens stimulating proliferation and maturation of lymphocytes.

TNFα is one of the best-studied proinflammatory cytokines secreted by innate immunity cells. TNFα binds to two types of transmembrane receptors, TNFR1 and TNFR2. Depending on the receptor type, this binding initiates signaling pathways regulating programmed cell death (apoptosis or necroptosis) and NF-kB activation, which leads to the induction of pro-survival signals [10]. TNFα-signaling in lymphocytes is related to immune modulation [11].

Among other regulators of the immune response, IFNγ and IL-2 are important cytokines associated with cytotoxic lymphocytes. Cytokines IFNγ and IL-2 stimulate the development of the immune response and support the functioning of effector cells. Thus, IFNγ-secreted lymphocytes, upon interaction with the target cell, enhance the expression of MHC class 1 in it, which makes such cells readily detectable to CD8^+^ cytotoxic lymphocytes [12]. IL-2 controls homeostasis and differentiation of T cells [13]. In particular, it stimulates the proliferation of naive CD8 lymphocytes and their maturation into cytotoxic lymphocytes capable of secreting IFNγ and killing infected cells [14]. Long-term incubation of lymphocytes with IL-2 induces their differentiation into LAK cells cytotoxic against tumor cells [14,15].

The purpose of this study was to analyze basic stages in the formation of cytotoxic lymphocyte subpopulations under the effect of Tag7 and the involvement of the aforementioned cytokines in these processes.

## 2. Materials and Methods

### 2.1. Cells

Human peripheral blood mononuclear cells (PBMCs) from healthy donors were isolated by sequential centrifugation in a Ficoll gradient as described [4]. All procedures performed were in accordance with the Declaration of Helsinki (1964) and its later amendments (World Medical Association, 2013) or comparable ethical standards and were approved by the medical ethics committee of FSBI N.N. Blokhin National Medical Research Center of Oncology of the Ministry of Health of the Russian Federation. Purified PBMCs (4 × 10^6^ cells/mL) were incubated in RPMI-1640 with 10% FCS and antibiotics (penicillin and streptomycin). Human erythroblastoma K562 cells used as targets for cytotoxicity assay were cultured in the same medium. This cell line was obtained from the cell line collection of N. N. Blokhin National Medical Research Center of Oncology of the Ministry of Health of Russia.

### 2.2. Magnetic Bead-Based Cell Separation

Lymphocyte subpopulations were isolated using Dynabeads Untouched kits (Invitrogen, USA) for human CD4, CD8 and NK cells (CD16^+^CD56^+^) and monocytes (CD14^+^). Antibodies to CD25 and CD132 were coupled to PanMouse IgG beads (Invitrogen, Waltham, MA, USA), and anti-CD122 antibodies were bounded to M-280 Sheep Anti-Rabbit IgG (Invitrogen, Waltham, MA, USA). All procedures were performed as recommended by the manufacturers.

### 2.3. Affinity Chromatography, Immunoadsorption and Western Blotting

sTREM-1 (AA 1-200, His tag, antibodies-online GmbH, Germany) was conjugated to CNBr-activated Sepharose 4B (GE, Chicago, IL, USA) according to the manufacturer’s protocol. The Tag7 was loaded onto the TREM-1-Sepharose 4B column. The column was thoroughly washed with PBS (contained 0.5 M NaCl) and PBS alone and then eluted with triethylamine (pH 12, 0.25 M). The eluted material was resolved by SDS-PAGE and blotted onto a nitrocellulose membrane. To detect Tag7, primary rabbit anti-Tag7 antibodies (1:10,000) followed by secondary HRP-conjugated anti-rabbit antibody (GE, Chicago, IL, USA; 1:15,000) were used. For visualization purposes, ECL Plus kit (GE, Chicago, IL, USA) was used according to the manufacturer’s protocol.

### 2.4. Activating Agents

Lymphocyte activation was induced by Tag7 and IL-2 or IFNγ added to a final concentration of 10^–9^ M. Recombinant Tag7 (PGLYRP1) was prepared as described [4]. Recombinant IL-2, TNFα and IFNγ were from Sigma (St. Louis, MO, USA).

### 2.5. Cytotoxicity Assay

Lymphocyte preparations were mixed with target K562 cells at a 20:1 ratio (1.2 × 10^6^ and 60 × 10^3^ cells, respectively), incubated at 37 °C in a 5% CO_2_ atmosphere for 3 h, and their cytotoxic activity was measured using a CytoTox 96 Assay kit (Promega, Madison, WI, USA) according to the manufacturer’s protocol.

### 2.6. Antibodies and Inhibitors

TREM-1 receptor activation was blocked by the inhibitory peptide LP17 (LQVTDSGLYRCVIYHPP, 10^−9^ M). Signaling molecules associated with the IL-2R receptor were blocked using specific inhibitors of JAK1 and JAK3 (5 µM) and of STAT3 and STAT5 (10 µM) (Santa Cruz Biotechnology, Santa Cruz, CA, USA). All inhibitors were added 1 h before lymphocyte treatment with an activating agent. Inhibitors of STAT3 and STAT5 were additionally added to PBMC on days 3 and 5 of incubation. Rabbit anti-Tag7 antibodies were obtained as described [4]. Mouse anti-human CD25 and rabbit anti-human CD122 antibodies were from Invitrogen (Waltham, MA, USA); mouse anti-human CD132 antibodies were from R&D (Minneapolis, MN, USA); mouse anti-human IFNγ, rat anti-human TNFα and rat anti-human IL-2 antibodies were from Invitrogen (Waltham, MA, USA).

### 2.7. ELISA

The levels of secretion of different cytokines were evaluated with Human IL-2, IFNγ and TNFα ELISA Kits (Thermo Fisher Scientific, Waltham, MA, USA) according to the manufacturer’s protocols.

### 2.8. qPCR

RNA was isolated from the fraction of CD3^+^ CD4^+^-lymphocytes (purified by magnetic cell separation) after their treatment with Tag7 (10^−9^ M) for 22 and 46 h. A sample of 4 × 10^6^ lymphocytes was lysed in 500 µL of Trizol Reagent (Invitrogen, Waltham, MA, USA), incubated at room temperature for 5 min, frozen and stored at −70 °C. RNA measurement was conducted with NanoDrop (Thermo scientific, USA), and equal amounts of RNA (1 μg) were used. For detection, RNA degradation electrophoresis was applied. The synthesis of cDNA was performed with oligo(dT) primers (Eurogen, Waltham, MA, Moscow, Russia). The products were used for qPCR with primers for genes encoding RPLP0, IL-2, CD25, CD122, CD132 and IFNγ. CD3^+^ CD4^+^-lymphocytes were purified from PBMC by magnetic bead isolation and then activated with TNFα and IFNγ. The level of RPLP0 mRNA was taken as a reference gene. The primers were as follows. RPLP0 forward: 5′-ACTGGAGACAAAGTGGGAGCC, reverse: 5′-CAGACACTGGCAACATTGCG; IL-2 forward: 5′ AAACTCACCAGGATGCTCAC 3′, reverse: 5′ TGTTTCAGATCCCTTTAGTTCCAG 3′; CD25 forward: 5′ CAGTTTCCAGGTGAAGAGAAGC 3′, reverse: 5′ CTGTTGTAAATATGGACGTCTCCA 3′; CD122 forward: 5′ CCTTTGAGAACCTTCGCCTG 3′, reverse: 5′ GGTGTCTTTCAAAGTAGTGGGAG 3′; CD132 forward: 5′ GCATTATTGGTACAAGAACTCGG 3′, reverse: 5′ ACAAATGTTTGGTAGAGGTGGA 3′; IFNγ forward: 5′ GGGTTCTCTTGGCTGTTACTG 3′; reverse: 5′ TTCTGTCACTCTCCTCTTTCCA 3′; TNFα forward: 5′-CTTCTCCTTCCTGATCGTGC-3′; reverse: 5′-GCTGGTTATCTCTCAGCTCCA-3′. Measurements at each point were made with at least three replications, and the mean value was calculated. Expression values were quantified using the equation of fold over control = 2^ΔΔCt^ method, where ΔCt represents the differences in cycle threshold numbers between the target gene and reference gene, and ΔΔCt represents the relative change in these differences between control and treatment groups.

### 2.9. Statistical Analysis

Data are presented as the average ± standard deviation. All experiments were repeated at least three times. Testing for significant differences between treatment and control was performed with MathCad software (PTC, Cambridge, MA, USA) using the Student t test for experiments on cell treatment with a single agent and 2-way ANOVA for experiments on cell treatment with 2 or more agents (see individual figure legends). *p*-values of less than 0.05 were considered significant.

## 3. Results

### 3.1. Treatment of PBMCs with Tag7 Results in the Formation of Cytotoxic Subpopulations of NK (CD16^+^CD56^+^), CD3^+^CD4^+^ and CD3^+^CD8^+^ Lymphocytes, But Only If the PBMC Pool Contains Monocytes

As we have previously shown, Tag7 can induce in PBMCs of at least three distinct subpopulations of cytotoxic lymphocytes that alternately exhibit their activity against HLA-negative tumor cell lines during a 6-day incubation period [7]. In this study, Tag7 also proved to induce the cytotoxic fraction of NK cells (CD16^+^CD56^+^), which reached a peak of activity on day 3–4. The number of these cells subsequently decreased significantly, and cytotoxicity associated with them was no longer observed (Appendix A). The cytotoxic activity of CD3^+^CD4^+^ lymphocytes was detected on days 4 and 6 of incubation with Tag7, while that of CD3^+^CD8^+^ lymphocytes was only detected on day 6 (Appendix A).

According to our previous data, Tag7 treatment of PBMCs depleted of monocytes does not result in the induction of cytotoxic activity [7]. Here, we tested whether monocytes are necessary for the formation of each cytotoxic lymphocyte subpopulation. Monocytes were removed from PBMCs by magnetic cell separation, and the resulting PBMC mc (–) cells were incubated with Tag7. Using the same method, NK cells and CD3^+^CD4^+^ lymphocytes were isolated from the Tag7-treated PBMC mc (–) cell pool on incubation day 4, and CD3^+^CD8^+^ lymphocytes were isolated on day 6. Cytotoxicity tests showed that the removal of monocytes prevented the induction of cytotoxic activity in either NK cells or CD3^+^CD4^+^ and CD3^+^CD8^+^ lymphocytes (Figure 1a). As a control, we also used an inhibitory peptide to TREM-1 receptor LP17, which was added to PBMC 1 h before activation Tag7. The data demonstrated the absence of the cytotoxic activity of PBMC during 6-day co-incubation of the LP17 peptide and Tag7 (Figure 1a). Thus, monocytes appear to be the first link in the chain of signal transmission from Tag7 to effector lymphocytes.

It is known that Tag7 is a ligand for the innate immune receptor TREM-1 [16]. We continued to study the interaction of Tag7 with TREM-1 by affinity chromatography. We detected the binding of a soluble form of TREM-1 immobilized on Sepharose with Tag7 (Figure 1b). An excess amount of Tag7 was passed through the column with TREM-1 immobilized on CNBr-Sepharose. Elution of Tag7 bound to TREM-1 was performed using triethylamine, and the material was analyzed by SDS-PAGE and WB. The elution material containing Tag7 was detected with specific antibodies (Figure 1b (1)). Recombinant Tag7 was used as a control for the obtained results, which were analyzed by SDS-PAGE and WB and developed with specific antibodies (Figure 1b (2)).

### 3.2. Tag7 Stimulates Secretion of Cytokines TNFα, IFNγ and IL-2

Taking into account that monocytes produce lymphocyte-activating factors [17], our next task was to analyze the profile of cytokines secreted to the medium by Tag7-activated PBMCs. First, PBMCs were incubated with Tag7 for 3 days, and samples of the conditioned medium were taken every 24 h for quantitative determination of proinflammatory cytokines TNFα и IFNγ by ELISA. As shown in Figure 2a, the level of TNFα reached a peak on day 2 and then decreased, while the level of IFNγ consistently increased during the incubation period. Thus, PBMCs treated with Tag7 secrete not only the proinflammatory cytokine TNFα but also IFNγ, which is known for its role in antiviral defense and the ability to activate lymphocytes, acting together with IL-2.

We then evaluated the profile of IL-2 secretion by Tag7-activated PBMCs and the involvement of monocytes in its induction. In this case, PBMCs were incubated with Tag7 for 6 days. In view of the data that TREM-1 activation may lead to the induction of genes coding for proinflammatory cytokines [7,9], the incubation was also performed in the presence of specific TREM-1 inhibitor LP17. Conditioned medium from untreated PBMC was used as additional control. The medium conditioned by PBMCs was sampled every 24 h. The results showed that the level of IL-2 consistently increased during the incubation period in both variants but was generally lower when TREM-1 activation was blocked or when untreated PBMC was used (Figure 2b). Despite the presence of a certain amount of IL-2 in conditioned medium, we did not observe cytotoxic activity after 6 days of co-incubation of LP17 and Tag7 with PBMC (Figure 1a). This is evidence that the interaction of Tag7 with TREM-1 is necessary for inducing PBMCs to produce and secrete a sufficient amount of IL-2 into the medium to a mature lymphocytes subpopulation.

### 3.3. CD3^+^CD4^+^ Lymphocytes Are the Main Source of IL-2 and Are Necessary for the Formation of Each Cytotoxic Subpopulation

The appearance of TNFα and IFNγ in the conditioned medium in the first days of Tag7 incubation with PBMC plays an important role in the formation of an activation signal. We hypothesized that these cytokines promote the activation of CD3^+^ CD4^+^-lymphocytes and the secretion of IL-2. In view of this hypothesis, we then analyzed the changes in the expression of mRNA of IL-2, IFNγ and TNFα in CD3^+^ CD4^+^-lymphocytes under the treatment of TNFα and IFNγ. A subpopulation of CD3^+^ CD4^+^-lymphocytes was isolated by magnetic separation from PBMCs and incubated with recombinant TNFα or IFNγ for 24 h. As shown in Figure 3a, incubation of CD3^+^ CD4^+^-lymphocytes with recombinant TNFα led to increased levels of IL-2 mRNA (45-fold), IFNγ (36-fold) and TNFα. Comparable results were detected in CD3^+^ CD4^+^ -lymphocytes incubated with IFNγ. The level of IL-2 mRNA increases greater (more than 97-fold) than IFNγ mRNA (15-fold) in the case of incubation of CD3^+^ CD4^+^-lymphocytes with recombinant IFNγ. In both cases, under the action of TNFα and IFNγ in lymphocytes, there is a slight change in TNFα mRNA expression by 2–4 folds. To prove the effect of TNFα and IFNγ cytokines on the secretion of IL-2 CD3^+^ CD4^+^ -lymphocytes, we analyzed the conditioned medium of pure CD3^+^ CD4^+^ -lymphocytes after TNFα or IFNγ treatment. We demonstrated that under TNFα or IFNγ treatment, CD3^+^CD4^+^-lymphocytes secrete IL-2 into conditioned medium (Appendix A). It was also shown that specific antibodies against cytokines TNFα, IFNγ and IL-2, which were co-incubated with Tag7-PBMC, decrease the cytotoxic effect of Tag7-PBMC against tumor cells (Appendix A).

Thus, we can point out that secretion of TNFα and IFNγ after incubation of Tag7 with PBMC in the first days triggers the activation of CD3^+^ CD4^+^-lymphocytes.

Since CD3^+^ CD4^+^-lymphocytes are the main source of IL-2 in the immune response system [18], it was relevant to assess the role of CD3^+^CD4^+^ cells in the transmission of activation signal from Tag7 and the production of IL-2 in the Tag7–PBMC system. Therefore, we evaluated IL-2 secretion in PBMCs depleted of CD3^+^ CD4^+^-lymphocytes by magnetic separation and incubated with Tag7 for 6 days. As shown in Figure 2b, no IL-2 was detected in the conditioned medium. Moreover, PBMCs depleted of CD3^+^ CD4^+^-lymphocytes showed no cytotoxic activity (Figure 3b). It appears that the cytotoxic activity of lymphocytes is stimulated by IL-2 produced by CD3^+^ CD4^+^-cells under the effect of Tag7 and secreted into the medium.

Our next task was to estimate the effect of IL-2 secreted into the medium by Tag7-treated PBMCs on the formation of individual cytotoxic lymphocyte subpopulations. To this end, subpopulations of NK (CD16^+^CD56^+^), CD3^+^CD4^+^ and CD3^+^CD8^+^ cells were isolated from PBMCs by magnetic separation and incubated with recombinant IL-2 for 4 or 6 days. The results showed that all these subpopulations acquired the ability to kill K562 tumor cells after treatment with IL-2. To test whether Tag7 directly induces cytotoxicity in effector cells, the same lymphocyte subpopulations were treated with this protein. No cytotoxic activity was detected in this case, indicating that Tag7 has no direct effect on this activity (Figure 3b). This is evidence that IL-2 produced by CD3^+^CD4^+^ lymphocytes plays a key role in inducing the cytotoxic activity of NK, CD3^+^CD4^+^ and CD3^+^CD8^+^ cells. 

We then analyzed changes in the expression of genes encoding IFNγ, IL-2 and IL-2 receptor subunits—IL-2Rα (CD25), IL-2Rβ (CD122) and IL-2Rγ (CD132)—in CD3^+^CD4^+^ lymphocytes under the effect of PBMC treatment with Tag7. PBMCs were incubated with Tag7 for 22 and 46 h, and the CD3^+^CD4^+^ subpopulation was then isolated by magnetic separation to evaluate changes in the levels of mRNAs from the above genes. After 22 h incubation, the mRNA levels for IL-2 and CD25 receptor subunit were enhanced most strongly (more than 130-fold), while enhancement in the mRNA levels for other receptor subunits and IFNγ was an order of magnitude lower, with a similar pattern being observed after 46 h (Figure 3c). Thus, CD3^+^CD4^+^ cells from the Tag-7-treated PBMC pool showed an enhanced gene expression for cytokines IL-2 and IFNγ, which stimulate lymphocyte activation, and for IL-2 receptor subunits (especially CD25), which improve IL-2 signal transmission, such that CD25 (IL-2Rα) ensures high affinity of IL-2 binding.

These results confirm that IL-2 produced by CD3^+^CD4^+^ cells plays a key role in inducing the cytotoxic activity of lymphocytes: if these cells are removed from the Tag7–PBMC system, IL-2 is absent in the conditioned medium, and active cytotoxic lymphocytes are not generated.

### 3.4. Role of IL-2 Receptor in the Induction of Cytotoxic Lymphocytes

Cells expressing individual IL-2R subunits were removed from PBMCs by magnetic separation, and the remaining cells were treated with Tag7 and tested for cytotoxic activity. As shown in Figure 4a, no cytotoxicity was observed after the removal of cells expressing IL-2Rα (CD25) or IL-2Rγ (CD132) subunits, whereas the removal of IL-2Rβ (CD122)-positive cells did not prevent the induction of cytotoxic lymphocytes. Therefore, two IL-2 receptor subunits, CD25 and CD132, are necessary for the formation of cytotoxic lymphocyte subpopulations.

Cytokine IL-2 secreted to the medium during PBMC incubation with Tag7 is the main factor responsible for activation of cytotoxic lymphocyte subpopulations. Therefore, it was relevant to estimate what stages of signal transduction from the IL-2 receptor are indispensable for the induction of lymphocyte cytotoxicity. It is known that cytoplasmic kinases JAK1 и JAK3 are functionally coupled to the IL-2 receptor and required for signal transduction [19]. To block their activity, we used specific JAK1 and JAK3 inhibitors, which were added to PBMCs 1 h before their treatment with Tag7. No cytotoxic effect was observed after 6 days of incubation with each of the inhibitors (Figure 4b), indicating that the induction of cytotoxicity depends on activation of JAK1 and JAK3 kinases upon IL-2 binding to the receptor. These kinases, in turn, activate STAT3 and STAT5 proteins, which are translocated as dimers into the nucleus and initiate gene transcription [20,21]. The role of this process in signal transduction was evaluated using STAT3 and STAT5 inhibitors that block their dimerization. The experimental procedure was the same as above, but the results showed that treatment with these inhibitors did not affect the induction of cytotoxic lymphocytes. Therefore, the IL-2–JAK signaling pathway does not depend on dimerization of STAT3 and STAT5 transcription factors.

## 4. Discussion

Here, we have described the basic stages of signal transmission from Tag7 to effector lymphocytes and identified the factors responsible for the induction of cytotoxic lymphocyte subpopulations capable of killing immune-evasive tumor cells.

The induction of such subpopulations under the effect of Tag7 depends on the consecutive activation of monocytes and CD3^+^ CD4^+^-lymphocytes. Tag7 binding to the TREM-1 receptor on the surfaces of monocytes stimulates the production of not only proinflammatory cytokine TNFα but also of IFNγ, which has been shown for the first time. We have shown that treating CD3^+^ CD4^+^-lymphocytes with TNFα and IFNγ leads to the expression of mRNA of activating cytokines IL-2 and IFNγ. Thus, the cytokines secreted by monocytes transmit the activation signal to CD3^+^ CD4^+^-lymphocytes, which in turn play a central role in its transmission to effector lymphocyte subpopulations.

Cytokine IL-2 is the main factor of T lymphocyte development [22]. Tag7-activated CD3^+^CD4^+^ lymphocytes are the main source of IL-2, which is necessary for the maturation of each subpopulation of cytotoxic CD16^+^CD56^+^, CD3^+^CD4^+^ and CD3^+^CD8^+^ cells. The CD16^+^CD56^+^-lymphocytes are activated by day 4 of Tag7 treatment, and then these cells disappear within a very short time [7]. We detected cytotoxic activity of CD3^+^CD4^+^-lymphocytes on the 4th and 6th days of incubation of PBMC with Tag7 and its absence on the 5th day. We suppose that on the 4th and 6^th^ days the cytotoxic activity of CD3^+^CD4^+^-lymphocytes was caused by different subpopulations of CD3^+^CD4^+^-lymphocytes, while the CD3^+^CD8^+^ lymphocyte population becomes active only by the 6th day of incubation PBMC with Tag7. It is possible that the maturation of CD3^+^CD8^+^-lymphocytes depends on the level of IL-2 cytokine, which continuously increases over 6 days.

IL-2 cytokine induces noncanonical cytotoxic T cell subpopulations that produce a nonspecific immune response against MHC-negative tumor cells [6]. We have shown here that the level of IL-2 in the medium conditioned by Tag7-treated PBMCs consistently increased during the 6-day incubation period. This may be explained by changes in gene expression in CD4^+^ lymphocytes that lead to enhanced mRNA synthesis for IL-2 and the subunits of its receptor. The expression of these subunits on CD4^+^ lymphocytes provides higher affinity of IL-2 binding and additional autostimulation of CD3^+^ CD4^+^-lymphocytes. The gene coding for IFNγ is also activated in CD3^+^ CD4^+^-lymphocytes. This factor apparently acts synergistically with IL-2 and contributes to lymphocyte activation.

The receptor for IL-2 on the cell surface consists of three subunits: IL-2Rα (CD25), IL-2Rβ (CD122) and IL-2Rγ (CD132) [22]. All of them are necessary for high-affinity binding of this cytokine. Our results show that at least two subunits—IL-2Rα and IL-2Rγ—should be expressed on the surface of lymphocytes to ensure IL-2 signal transmission and induction of cytotoxicity; i.e., they are sufficient for the assembly of a high-affinity receptor that allows lymphocytes to receive the activation signal.

The interaction of IL-2 with its receptor induces activation signal transduction via JAK1 and JAK3 tyrosine kinases associated with the β and γ subunits of the receptor, respectively [19]. Our results show that the induction of lymphocyte occurs via this signal transduction pathway, with the involvement of JAK1 and JAK3 kinases, but does not depend on dimerization of STAT3 and STAT5 transcription factors. It may well be that inhibitors used in this study prevent dimerization but do not interfere with the functional activity of these factors, since there is published evidence for the involvement of STAT proteins in signal transduction from the IL-2 receptor [20,21].

Thus, a probable scheme of activation signal transmission from Tag7 to effector lymphocytes in PBMCs is as follows (Scheme 1). First, Tag7 binds to TREM-1 receptor on monocytes and thereby activates the genes of proinflammatory cytokines. Cytokines secreted in the early days, TNFα and IFNγ, influence CD3^+^ CD4^+^ lymphocytes. This stage is essential for the activation of CD3^+^ CD4^+^-lymphocytes, which play a central role in subsequent regulation. These lymphocytes are the source of IL-2, which is necessary for the maturation of cytotoxic subpopulations of NK cells and CD3^+^CD8^+^ lymphocytes and also stimulates the cytotoxicity of the CD3^+^CD4^+^ subpopulation. The efficiency of the signal received from IL-2 depends on the expression of all receptor subunits on the cell surface, which ensures the assembly of the cytokine–high-affinity-receptor complex and signal transduction in the cell.

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
