# Peer review of "Cytokines TNFα, IFNγ and IL-2 Are Responsible for Signal Transmission from the Innate Immunity Protein Tag7 (PGLYRP1) to Cytotoxic Effector Lymphocytes"

_cells, 2020, doi:10.3390/cells9122602_

Round 1
Reviewer 1 Report
The aim of authors is interestingly, as they examine the molecular and cellular mechanisms that trigger the regarding the clarification in detail the mechanism of transmission of the Tag7 signal in the activation of an antimicrobial defense in mammalian. In fact, Tag7 (PGLYRP1) protein is one factor belonging to the family of peptidoglycan recognition proteins (PGRPs) encoded by evolutionary conserved genes involved in the innate and adaptive immune response for defense against alien agents.
The authors, with an in vitro model would provide further information on the involvement of Tag7 in signal transmission to the TLR4 receptor and thereby activating the production of antimicrobial peptides, by analyzing basic stages in the formation of cytotoxic lymphocyte subpopulations under the effect of Tag7 and the involvement of TNFa and INFg cytokines in these processes. These studies could contribute to the development of new means for immunotherapy.
The topic of the manuscript is inherent to the aim and scope of the journal “Cells”.
The design of the study is well done, and the paper clearly written. The experiments are well conducted and organized in the aims of authors, both from an experimental point of view and in terms of the techniques used to confirm the objective to be achieved. The results are enough well discussed. The conclusions are appropriate.
However, some minor correction should be do.
- IN THE “MATERIALS AND METHODS” SECTION:
Subchapter: Cells
Line 91: probably missing the space between the point after “(pH 12, 0.25 M)” and "The". The authors should add it.
Subchapter: Quantitative real-Time
Line 124: The authors wrote "control". Authors should write "houskeeping" more correctly.
Moreover, in this subchapter the authors should detail better the protocol of RT –qPCR.
- The authors should add which RNA integrity has been used for Real-Time PCR analyses.
- The authors should add which quantity of RNA has been reverse transcribed and then how much has been used for quantitative real time PCR.
- The authors should detail better what kind of analysis they used for qPCR.
- As for the primers used, did the authors perform amplification efficiency curves? What is the efficiency threshold resulted and considered for each primer? The authors should be specify.
- The authors used only RPLP0 as internal reference. I suggest to do analysis according to the MIQE guidelines (Bustin, S.A., Benes, V., Garson, J.A., Hellemans, J., Huggett, J., Kubista, M. et al. (2009) The MIQE guidelines: minimum information for publication of quantitative Real-Time PCR experiments. Chem. 55, 611–622, https://doi.org/10.1373/clinchem.2008.112797), and then add housekeepings.
Subchapter: Statistical Analysis
This section is missing references to statistical analyses about Student t test. Since in the legends of Figures 1 and S1 the Student t test appears, the authors must add and specify the use of this test in this subchapter.
Morover, the authors must indicate and specify what level of significance is set and thus which p-value was considered as the limit for the significance of the result.
- IN THE “RESULTS” SECTION:
- In the results comments there is a complete lack of a description of statistical analysis. The authors should in all paragraphs of the results added after each conclusion the value of the data expressed as an means ± SD with relative p-value has been evaluated as significant.
Subchapter: 3.1 Treatment of PBMCs with Tag7 results in the formation of cytotoxic subpopulations of NK (CD16+CD56+), CD3+CD4+, and CD3+CD8+ lymphocytes, but only if the PBMC pool contains monocytes.
Line 148-150: The authors wrote: “In this study, Tag7 also proved to induce the cytotoxic fraction of NK cells (CD16+CD56+), which reached a peak of activity on day 3–4. The number of these cells subsequently decreased significantly, and cytotoxicity associated with them was no longer observed (Suppl. Fig. S1). The cytotoxic activity of CD3+CD4+ lymphocytes was detected on days 4 and 6 of incubation with Tag7, while that of CD3+CD8+ lymphocytes, only on day 6 (Suppl. Fig. S1)”. The authors does not provide description of the results obtained at day 5. From the results obtained and shown in the graph of Figure S1 it is possible to see a significant decrease in activity for all three different populations at day 5. How do the authors explain this? What does it mean? The authors must add and comment also on this data/result.
Line 161/162: The authors wrote: "We detected the binding of a soluble form of TREM-1 immobilized on Sepharose with Tag7." Is not indicated where TREM-1 was recovered from. Authors should specify and detail better the experiment protocol to the regard of where it was recovered from, and then comment the results.
Line 176-178: The authors wrote: "The results showed that the level of IL-2 consistently increased during the incubation period in both variants but was generally lower when TREM-1 activation was blocked (Fig. 2В)”. There are no correlation graphs and /or comment in the work in which a positive or negative association are highlighted to confirm these conclusions. I suggest to the authors to validate the existence of a correlation to provide a more consistent results.
Moreover, the decrease in IL2 secretion highlighted by blocking TREM-1 on monocytes by inhibitory peptide LP17 (10–9 M) added 1 h before incubation with Tag7 (inhTREM-1) compared to that obtained by incubating PBMCs only with Tag7, was it significant? The authors should indicate and comment on this result.
Subchapter: 3.3 CD3+CD4+ lymphocytes are the main source of IL-2 and are necessary for the formation of each cytotoxic subpopulation
Line 188-193: The authors wrote: "As shown in Fig. 3A, incubation of CD3+ CD4+-lymphocytes with recombinant TNFα leads to increase level of IL-2 mRNA (45-fold), IFNγ (36-fold) and TNFα. Comparable results were detected in CD3+ CD4+ -lymphocytes incubated with IFNγ. The level of IL-2 mRNA increases greater (more than 97-fold) than IFNγ mRNA (15-fold) in case of incubation CD3+ CD4+-lymphocytes with recombinant IFNγ. In both cases, under the action of TNFα and IFNγ in lymphocytes, there is a slight change in TNFα mRNA expression: 2-4 times.” The authors indicate only how much the expression of cytokines of interest in the study increases compared to control (x-fold!), and do not specify whether these increases are significant. The authors should here, as well as in all other subchapters of the results, indicate if and how statistically significant are the results obtained to consolidate the data, indicating specific means ± SD.
Moreover, even in this case, are the obtained data of cytokine up-regulation confirmed by proteins? For example by ELISA. I think the authors should do it and present it to make the data more complete.
- FIGURES
With regard to the figures, the graphs are clear, but no statistical results are presented. There is only a summary description in the legend ("Differences from the control in all cases are significant at p<0.05“). The authors should be specify and add in the various graphs the statistical indications (i.e. *, **, ...) with the description of “*, **, …” and the relative p obtained for each sample in the legend.
- IN THE “FIGURE LEGENDS” SECTION:
Legend of Figure 1.
Line 257: The authors wrote: “(Student t test)”. In the “Statistical analysis section is missing references to this kind of statistical analysis. If the authors used this test specify better both in the text and here in legend and make it clear and visible in the graph the statistical analysis results.
Line 257-258: The authors wrote: “(b) Western blot analysis of Tag7, applied to the TREM-1 conjugated Sepharose column (1) and then excessively washed with PBS and Tag7 as a control (2)”. In the “Materials and Methods” section is missing references to Western Blot. Ther is olny indication of “Affinity Chromatography, Immunoadsorption, and Immunoblotting”. The authors should use the same terminology, correct. Moreover, it is not clear what is in column (1) and what is in column (2). The authors should better describe this part, indicating also what correspond at 20kDa in the SDS-PAGE shown.
Legend of Figure 3.
Line 273: the authors should replace “RPLP” with “RPLP0”
- IN THE “DISCUSSION” SECTION:
Line 329: To the regard: “probable scheme”. I would like to suggest to the authors to add a figure presenting a summary of the pathway of activation signal transmission from Tag7 to effector lymphocytes in PBMCs. The authors should clearly present in this figure the involvement of all the different lymphocyte subpopulations induced in PBMC incubated with Tag7, their relation with cytokines, with with activators and inhibitors, related to the results obtained.

Author Response
- IN THE “MATERIALS AND METHODS” SECTION:
Subchapter: Cells
Line 91: probably missing the space between the point after “(pH 12, 0.25 M)” and "The". The authors should add it.
Missing space was added line 91.
Subchapter: Quantitative real-Time
Line 124: The authors wrote "control". Authors should write "houskeeping" more correctly.
Moreover, in this subchapter the authors should detail better the protocol of RT –qPCR.
- The authors should add which RNA integrity has been used for Real-Time PCR analyses.
- The authors should add which quantity of RNA has been reverse transcribed and then how much has been used for quantitative real time PCR.
- The authors should detail better what kind of analysis they used for qPCR.
- As for the primers used, did the authors perform amplification efficiency curves? What is the efficiency threshold resulted and considered for each primer? The authors should be specify.
- The authors used only RPLP0 as internal reference. I suggest to do analysis according to the MIQE guidelines (Bustin, S.A., Benes, V., Garson, J.A., Hellemans, J., Huggett, J., Kubista, M. et al. (2009) The MIQE guidelines: minimum information for publication of quantitative Real-Time PCR experiments. Chem. 55, 611–622, https://doi.org/10.1373/clinchem.2008.112797), and then add housekeepings.
Additional data were added to
Line 126-127: “RNA measurement was conducted with NanoDrop (Thermo scientific, USA) and equal amounts of RNA (1 μg) were used. For detection RNA degradation electrophoresis was applied’
Line 140-144: “Measurements at each point were made in at least three replications, and the mean value was calculated. Expression values were quantified using the equation: fold over control = 2ΔΔCt method, where ΔCt represents the differences in cycle threshold numbers between the target gene and reference gene, and ΔΔCt represents the relative change in these differences between control and treatment groups”.
Subchapter: Statistical Analysis
This section is missing references to statistical analyses about Student t test. Since in the legends of Figures 1 and S1 the Student t test appears, the authors must add and specify the use of this test in this subchapter.
Morover, the authors must indicate and specify what level of significance is set and thus which p-value was considered as the limit for the significance of the result.
We added references to to statistical analyses about Student t test at line 148-150: “Testing for significant differences between treatment and control was performed with MathCad software (PTC, Cambridge, MA) using the Student t test for experiments on cell treatment with a single agent and 2-way ANOVA for experiments on cell treatment with 2 or more agents (see individual figure legends)”.
We also have corrected figures and its legends and added level of significance in the each of experiments.
- IN THE “RESULTS” SECTION:
- In the results comments there is a complete lack of a description of statistical analysis. The authors should in all paragraphs of the results added after each conclusion the value of the data expressed as an means ± SD with relative p-value has been evaluated as significant.
We have corrected figures and its legend and added level of significance in the each of experiments.
“Line 148-150: The authors wrote: “In this study, Tag7 also proved to induce the cytotoxic fraction of NK cells (CD16+CD56+), which reached a peak of activity on day 3–4. The number of these cells subsequently decreased significantly, and cytotoxicity associated with them was no longer observed (Suppl. Fig. S1). The cytotoxic activity of CD3+CD4+ lymphocytes was detected on days 4 and 6 of incubation with Tag7, while that of CD3+CD8+ lymphocytes, only on day 6 (Suppl. Fig. S1)”. The authors does not provide description of the results obtained at day 5. From the results obtained and shown in the graph of Figure S1 it is possible to see a significant decrease in activity for all three different populations at day 5. How do the authors explain this? What does it mean? The authors must add and comment also on this data/result”
We have previously shown and here we have confirmed that after 5 days of incubation of PBMC with Tag7, no cytotoxic activity was detected (Sharapova, T.N.; Ivanova, O.K.; Soshnikova, N.V.; Romanova, E.A.; Sashchenko, L.P.; Yashin, D.V. Innate Immunity Protein Tag7 Induces 3 Distinct Populations of Cytotoxic Cells That Use Different Mechanisms to Exhibit Their Antitumor Activity on Human Leukocyte Antigen-Deficient Cancer Cells. J Innate Immun 2017, doi:10.1159/000479382). The lack of cytotoxic activity on day 5 may be due to several factors:
- The number of CD16+CD56+- cells reaches a maximum on the 3rd day, after which it decreases. Thus after 4 days of incubation, we detect only subpopulations of T lymphocytes (Data were published in Sharapova, T.N.; Ivanova, O.K.; Soshnikova, N.V.; Romanova, E.A.; Sashchenko, L.P.; Yashin, D.V. Innate Immunity Protein Tag7 Induces 3 Distinct Populations of Cytotoxic Cells That Use Different Mechanisms to Exhibit Their Antitumor Activity on Human Leukocyte Antigen-Deficient Cancer Cells. J Innate Immun 2017, doi:10.1159/000479382).
- We do not observe cytotoxic activity of CD3+CD8+-lymphocytes during 5 days of incubation. We suppose that the maturation of CD3+CD8+-lymphocytes depends on the level of IL-2 cytokine, which continuously increases over 6 days. The amount of cytokines secreted in the first days of incubation PBMC with Tag7 is insufficient for the formation of cytotoxic CD3+CD8+-subpopulations. Apparently, the formation of cytotoxic CD3+CD8+-lymphocytes takes longer.
It is quite possible that another cytokine should appear in the conditioned medium for the maturation of cytotoxic CD3+CD8+-lymphocytes. Our unpublished data indicate that after 3 days of incubation Tah7 with PBMC, IL-15 appears in a conditioned medium. On the other point due to co-activation CD3+CD8+-lymphocytes of IL-2 with IL-15 we detect increasing cytotoxic activity compared with only IL-2 activated of CD3+CD8+-lymphocytes.
- We detect the cytotoxic activity of CD3+CD4+ - lymphocytes on the 4th and 6th days of incubation of PBMC with Tag7 with an absence on the 5th day. We suppose that on the 4th and 6th the cytotoxic activity of CD3+CD4+ - lymphocytes are caused by different subpopulations of CD3+CD4+ -lymphocytes. However, this issue requires further research.
Comment in article: Discussion LINE 346-353
“Line 161/162: The authors wrote: "We detected the binding of a soluble form of TREM-1 immobilized on Sepharose with Tag7." Is not indicated where TREM-1 was recovered from. Authors should specify and detail better the experiment protocol to the regard of where it was recovered from, and then comment the results.”
Detailed data were inserted in article in line 178-183.
An excess amount of Tag7 was passed through the column with TREM-1 immobilized on CNBr-Sepharose. Elution of Tag7 bound to TREM-1 was performed using triethylamine and the material was analyzed by SDS-PAGE and WB. The elution material containing Tag7 was detected with specific antibodies (Fig. 1B-(1)). Recombinant Tag7 was used as a control for the obtained results, which were analyzed by SDS-PAGE and WB and developed with specific antibodies (Fig. 1B- (2)).
“Line 176-178: The authors wrote: "The results showed that the level of IL-2 consistently increased during the incubation period in both variants but was generally lower when TREM-1 activation was blocked (Fig. 2В)”. There are no correlation graphs and /or comment in the work in which a positive or negative association are highlighted to confirm these conclusions. I suggest to the authors to validate the existence of a correlation to provide a more consistent results.
We have corrected figures and its legend and added level of significance in the each of experiments.
Moreover, the decrease in IL2 secretion highlighted by blocking TREM-1 on monocytes by inhibitory peptide LP17 (10–9 M) added 1 h before incubation with Tag7 (inhTREM-1) compared to that obtained by incubating PBMCs only with Tag7, was it significant? The authors should indicate and comment on this result.”
Despite the secretion of IL-2 even in the presence of the inhibitory peptide TREM-1, we do not observe cytotoxic activity in PBMC. In this article we insert that data as control in (Fig. 1A) and comment at line 171-174.
Subchapter: 3.3 CD3+CD4+ lymphocytes are the main source of IL-2 and are necessary for the formation of each cytotoxic subpopulation
Line 188-193: The authors wrote: "As shown in Fig. 3A, incubation of CD3+ CD4+-lymphocytes with recombinant TNFα leads to increase level of IL-2 mRNA (45-fold), IFNγ (36-fold) and TNFα. Comparable results were detected in CD3+ CD4+ -lymphocytes incubated with IFNγ. The level of IL-2 mRNA increases greater (more than 97-fold) than IFNγ mRNA (15-fold) in case of incubation CD3+ CD4+-lymphocytes with recombinant IFNγ. In both cases, under the action of TNFα and IFNγ in lymphocytes, there is a slight change in TNFα mRNA expression: 2-4 times.” The authors indicate only how much the expression of cytokines of interest in the study increases compared to control (x-fold!), and do not specify whether these increases are significant. The authors should here, as well as in all other subchapters of the results, indicate if and how statistically significant are the results obtained to consolidate the data, indicating specific means ± SD.
Statistical significance was added to each comment of figure.
Moreover, even in this case, are the obtained data of cytokine up-regulation confirmed by proteins? For example by ELISA. I think the authors should do it and present it to make the data more complete.
In additional experiments we also investigate secretion of IL-2 by CD3+CD4+ - lymphocytes activated TNFα and IFNγ. We have showed that under TNFα or IFNγ treatment, CD3+CD4+ - lymphocytes secrete IL-2 to conditioned medium. These data confirm the point that CD3+CD4+ - lymphocytes are activated under the action of TNFα and IFNγ cytokines (Suppl. Fig.S2).
Comments were added in the article at line 219-222.
- FIGURES
With regard to the figures, the graphs are clear, but no statistical results are presented. There is only a summary description in the legend ("Differences from the control in all cases are significant at p<0.05“). The authors should be specify and add in the various graphs the statistical indications (i.e. *, **, ...) with the description of “*, **, …” and the relative p obtained for each sample in the legend.
Statistical significance and decription were added to each comment of figure.
- IN THE “FIGURE LEGENDS” SECTION:
Legend of Figure 1.
Line 257: The authors wrote: “(Student t test)”. In the “Statistical analysis section is missing references to this kind of statistical analysis. If the authors used this test specify better both in the text and here in legend and make it clear and visible in the graph the statistical analysis results.
Additional corrections were added to legend of Figure 1 and in article Line 148-150.
Line 257-258: The authors wrote: “(b) Western blot analysis of Tag7, applied to the TREM-1 conjugated Sepharose column (1) and then excessively washed with PBS and Tag7 as a control (2)”. In the “Materials and Methods” section is missing references to Western Blot. Ther is olny indication of “Affinity Chromatography, Immunoadsorption, and Immunoblotting”. The authors should use the same terminology, correct. Moreover, it is not clear what is in column (1) and what is in column (2). The authors should better describe this part, indicating also what correspond at 20kDa in the SDS-PAGE shown.
Corrections were added to legend at Line 290-292: “Tag7, applied to the TREM-1 conjugated Sepharose column and then eluted and detected by WB (1). WB of Tag7 (20kDa) was used as a control (2).”
Legend of Figure 3.
Line 273: the authors should replace “RPLP” with “RPLP0”
Corrections were added at line 306 and 313.
- IN THE “DISCUSSION” SECTION:
Line 329: To the regard: “probable scheme”. I would like to suggest to the authors to add a figure presenting a summary of the pathway of activation signal transmission from Tag7 to effector lymphocytes in PBMCs. The authors should clearly present in this figure the involvement of all the different lymphocyte subpopulations induced in PBMC incubated with Tag7, their relation with cytokines, with with activators and inhibitors, related to the results obtained.
The scheme describing signal transmission from Tag7 to effector lymphocytes has been added to “discussion section” at line 377 and also as additional Figure at line 328-331.
Reviewer 2 Report
The authors tried to delineate the consequences of Tag7 stimulation of PBMCs to the final outcomes of development of cytotoxic effector lymphocytes, NK cells, CD8+ cells and CD4+ cells. They showed that cytokines produced by monocytes stimulated with Tag7 at early times and IL-2 from stimulated CD4 T cells are responsible for the development of cytotoxic cell activity against K562 cells.
However, their data are not so convincing from the following points.
The concentrations of TNFa and IFNg produced by PBMCs that had been stimulated with Tag7 for two or three days are in the range of 200 to 400 pg/ml. To test the roles of these cytokines, they used cytokines at 1 nM, around 20 ng/ml. It is possible that other cytokines produced by stimulated monocytes may have important roles. To prove the effect of such cytokines, other methods, such as cytokine depletion using specific antibodies, should be used. Fig.2b showed inefficient inhibition of IL-2 production by LP17 at 1nM. The authors should show the effectiveness of this concentration of LP17, and may discuss the TREM1-independent action of Tag7. In Fig. 3A, did they use IL-2 for stimulation? They need to show that the rapid induction of CD25 and IL-2 in response to Tag7 stimulation is really dependent on TNFa and IFNg. Controls are missing in Fig. 4A and Fig.4B. The authors indicated that both STAT3 and STAT5 are not involved in the Tag7-induced cytotoxic activity of PBMCs. They need to show the effectiveness of these STAT3 and STAT5 inhibitors during the time courses of several days.
Author Response
The authors tried to delineate the consequences of Tag7 stimulation of PBMCs to the final outcomes of development of cytotoxic effector lymphocytes, NK cells, CD8+ cells and CD4+ cells. They showed that cytokines produced by monocytes stimulated with Tag7 at early times and IL-2 from stimulated CD4 T cells are responsible for the development of cytotoxic cell activity against K562 cells.
However, their data are not so convincing from the following points.
“The concentrations of TNFa and IFNg produced by PBMCs that had been stimulated with Tag7 for two or three days are in the range of 200 to 400 pg/ml. To test the roles of these cytokines, they used cytokines at 1 nM, around 20 ng/ml. It is possible that other cytokines produced by stimulated monocytes may have important roles. To prove the effect of such cytokines, other methods, such as cytokine depletion using specific antibodies, should be used.”
Thank you for your accurate remark on the applied concentrations of cytokines for the activation of lymphocytes. Indeed, the concentration of TNFα and IFNγ cytokines that was used to activate CD4 lymphocytes was higher than the amount of cytokines detected by the ELISA. This is primarily due to the fact that in this experiment we work in purified subpopulations of CD3+CD4+ - lymphocytes and use recombinant proteins (cytokines). Additional experiments that prove role of that cytokines were also conducted. To estimate the effect of the selected cytokines, specific antibodies to TNFα, IFNγ and IL-2 were used. Each of antibody was added 1 hour before the activation of PBMC Tag7. As a result, we observed the absence of cytotoxic activity in case of application IL-2 antibodies. It may indicate the key role of IL2 for the development of effector lymphocytes (Suppl.Fig.S3).It was found decrease in cytotoxic activity by 60% in the case of incubation of Tag7-PBMC with antibodies to TNFα, in tests with antibodies to IFNγ, cytotoxic activity was reduced by 40%.
We have inserted the data in the article: Line-222-224. Supplemental Figure S3 was also added.
On the other side we also investigated secretion of IL-2 by CD3+CD4+ - lymphocytes activated TNFα and IFNγ. We have showed that under TNFα or IFNγ treatment, CD3+CD4+ - lymphocytes secrete IL-2 to conditioned medium. These data confirm the point that CD3+CD4+ - lymphocytes are activated under the action of TNFα and IFNγ cytokines (Suppl. Fig.S2).
Comments were added in the article at line 219-222. Supplemental Figure S2 was also added.
We don’t deny important role other cytokines on development effector cell under the treatment of Tag7 but this issue requires further research.
“Fig.2b showed inefficient inhibition of IL-2 production by LP17 at 1nM. The authors should show the effectiveness of this concentration of LP17, and may discuss the TREM1-independent action of Tag7.”
As we use PBMCs from anonymous healthy donors, however, we do not know their immune status. It is possible that inactivated CD3+CD4+ - lymphocytes constitutively secrete a certain level of cytokines, in particular, IL-2. Therefore we investigated IL-2 secretion by PBMC through 6-day incubation without any additional treatment. We have demonstrated that through 6 –day incubation PBMC with RPMI-1640 + 10%FCS, lymphocytes secrete IL-2 in conditioned medium on a level comparable to the case with LP17+Tag7 treatment. Nevertheless, IL-2 secretion in both cases didn’t induce appearance of cytotoxic lymphocytes. Thus we did not observe TREM-1 independent action of Tag7 on IL-2 secretion.
We have added corrections at line 197-204 and inserted control data to Fig.2B.
We used the same concentration LP17 in experiment with IL-2 secretion as we used in experiments to determine cytotoxicity.
In this article we insert that data as control in (Fig. 1A) and comment at line 171-174.
“In Fig. 3A, did they use IL-2 for stimulation? They need to show that the rapid induction of CD25 and IL-2 in response to Tag7 stimulation is really dependent on TNFa and IFNg. Controls are missing in Fig. 4A and Fig.4B.”.
We are very grateful to the reviewer for carefully reading the article and apologize for undesignated data. In Fig.3A we didn’t use IL-2 for stimulation, in each case we used only TNFα or IFNγ. Corrections were also added to description of Fig.3A. Corrections (missed controls) were added to Fig. 4A and Fig.4B.
“The authors indicated that both STAT3 and STAT5 are not involved in the Tag7-induced cytotoxic activity of PBMCs. They need to show the effectiveness of these STAT3 and STAT5 inhibitors during the time courses of several days”.
We have conducted additional experiments where STAT3 or STAT5 inhibitors (same amounts) added on 1 (before Tag7), 3 and 5 days PBMC incubation and didn’t detect changes in cytotoxicity compare with our previous results. To be more objective we have corrected protocol of experiments in legend of Fig.4B (line 323-324) and in section “Antibodies and inhibitors” (line 114-115).
Round 2
Reviewer 2 Report
The authors responded well to my review comments. I do not see any problems.